Dairy management practices associated with multi-drug resistant fecal commensals and Salmonella in cull cows: a machine learning approach

http://orcid.org/0000-0001-7649-0649 Pandit Pranav S. 1
http://orcid.org/0000-0002-7534-732X Williams Deniece R. 2
Rossitto Paul 2
Adaska John M. 3
http://orcid.org/0000-0003-2028-8761 Pereira Richard 4
http://orcid.org/0000-0003-0896-3939 Lehenbauer Terry W. 2 4
http://orcid.org/0000-0002-8907-1272 Byrne Barbara A. 5
Li Xunde 6
Atwill Edward R. 6
http://orcid.org/0000-0003-0330-5013 Aly Sharif S. 2 4 saly@ucdavis.edu
1 EpiCenter for Disease Dynamics, One Health Institute, School of Veterinary Medicine, University of California Davis , Davis, CA , The United States of America
2 Veterinary Medicine Teaching and Research Center, University of California, Davis , Tulare, CA , The United States of America
3 California Animal Health and Food Safety Laboratory System, University of California, Davis , Tulare, CA , The United States of America
4 Department of Population Health and Reproduction, School of Veterinary Medicine, University of California, Davis , Davis, CA , The United States of America
5 Department of Pathology, Microbiology & Immunology, School of Veterinary Medicine, University of California , Davis, CA , The United States of America
6 Western Institute for Food Safety and Security, University of California, Davis , Davis, CA , The United States of America
Gillespie Joseph
Electronic publication date: 2021 Jul 16
Publication date: 2021
Volume: 9
Electronic Location ID: e11732
Received 2021 Jan 13; Accepted 2021 Jun 16
Copyright: © 2021 Pandit et al.
Copyright year: 2021
Copyright holder: Pandit et al.
License: This is an open access article distributed under the terms of the Creative Commons Attribution License, which permits unrestricted use, distribution, reproduction and adaptation in any medium and for any purpose provided that it is properly attributed. For attribution, the original author(s), title, publication source (PeerJ) and either DOI or URL of the article must be cited.
License URL: https://creativecommons.org/licenses/by/4.0/

Keywords: Dairy cattle, Cull cows, Antimicrobial resistance, Random forest, Gradient boosting, Decision tree classification, Salmonella, Enterococcus, Escherichia coli

Funding: U.S. Department of Agriculture CA-V-PHR-4707-AH407 and CA-V-PHR-4048-H USDA National Institute of Food and Agriculture (NIFA) 201567030238928 Funding for this research was provided by the U.S. Department of Agriculture Project number CA-V-PHR-4707-AH407 and CA-V-PHR-4048-H. This study was also supported by the USDA National Institute of Food and Agriculture (NIFA) project number 201567030238928. There was no additional external funding received for this study. The funders had no role in study design, data collection and analysis, decision to publish, or preparation of the manuscript.

==============================
Background

Understanding the effects of herd management practices on the prevalence of multidrug-resistant pathogenic Salmonella and commensals Enterococcus spp. and Escherichia coli in dairy cattle is key in reducing antibacterial resistant infections in humans originating from food animals. Our objective was to explore the herd and cow level features associated with the multi-drug resistant, and resistance phenotypes shared between Salmonella, E. coli and Enterococcus spp. using machine learning algorithms.

Methods

Randomly collected fecal samples from cull dairy cows from six dairy farms in central California were tested for multi-drug resistance phenotypes of Salmonella, E. coli and Enterococcus spp. Using data on herd management practices collected from a questionnaire, we built three machine learning algorithms (decision tree classifier, random forest, and gradient boosting decision trees) to predict the cows shedding multidrug-resistant Salmonella and commensal bacteria.

Results

The decision tree classifier identified rolling herd average milk production as an important feature for predicting fecal shedding of multi-drug resistance in Salmonella or commensal bacteria. The number of culled animals, monthly culling frequency and percentage, herd size, and proportion of Holstein cows in the herd were found to be influential herd characteristics predicting fecal shedding of multidrug-resistant phenotypes based on random forest models for Salmonella and commensal bacteria. Gradient boosting models showed that higher culling frequency and monthly culling percentages were associated with fecal shedding of multidrug resistant Salmonella or commensal bacteria. In contrast, an overall increase in the number of culled animals on a culling day showed a negative trend with classifying a cow as shedding multidrug-resistant bacteria. Increasing rolling herd average milk production and spring season were positively associated with fecal shedding of multidrug- resistant Salmonella. Only six individual cows were detected sharing tetracycline resistance phenotypes between Salmonella and either of the commensal bacteria.

Discussion

Percent culled and culling rate reflect the increase in culling over time adjusting for herd size and were associated with shedding multidrug resistant bacteria. In contrast, number culled was negatively associated with shedding multidrug resistant bacteria which may reflect producer decisions to prioritize the culling of otherwise healthy but low-producing cows based on milk or beef prices (with respect to dairy beef), amongst other factors. Using a data-driven suite of machine learning algorithms we identified generalizable and distant associations between antimicrobial resistance in Salmonella and fecal commensal bacteria, that can help develop a producer-friendly and data-informed risk assessment tool to reduce shedding of multidrug-resistant bacteria in cull dairy cows.

Introduction

The Centers for Disease Control and Prevention (CDC) estimates that more than 2.8 million antimicrobial resistant infections occur in the U.S. with more than 35,000 deaths annually (Control CfD, & Prevention, 2019). Amongst the resistant bacteria, the CDC classifies nontyphoidal Salmonella enterica as a serious public health threat (Control CfD, & Prevention, 2019). Salmonella is an important foodborne zoonotic agent in the U.S. (Scallan et al., 2011) and several studies reported on its prevalence in cull cattle. Troutt et al. found that the prevalence of Salmonella in cecal contents of dairy cattle at slaughter in the Western U.S. ranged between 9.6% and 35.6% in the winter, and between 32.3% and 93% in the summer (Troutt et al., 2001). More recent studies reported on the associations between herd management and seasonal differences on the prevalence of Salmonella in fecal samples of cull dairy cows collected quarterly from seven California dairies with an overall Salmonella shedding prevalence of 3.42% (95% CI [1.28–5.56]) (Abu Aboud et al., 2016). Pereira et al. (2019) followed six of the same California dairies for a second year showing an increase in Salmonella shedding prevalence (30.6%; 95% CI [262–350]). The increase in the prevalence was speculated to be due to increased rainfall and drier summer season and herd changes that occurred during the latter study (Adaska et al., 2020). The latter study explored how the study herd and cow level features were associated with shedding of antimicrobial resistant Salmonella and although 60% of the isolates were pan-susceptible, the remaining isolates were found to be resistant to different medically important antimicrobial drugs (MIAD) defined as antimicrobial drugs (AMD) that are important for therapeutic use in humans, with 12% of the isolates being multidrug resistance to more than two drug classes.

Fecal commensals such as E. coli and Enterococcus spp. can acquire mobile gene elements that encode antimicrobial resistance to these species (Aidara-Kane et al., 2018). Given the documented antimicrobial resistance in Salmonella isolated from cull dairy cows, further research on the similarity between resistance patterns of fecal commensals and Salmonella shed in feces of dairy cattle is needed.

Traditional risk factor approaches (mixed-effects modeling) can often have limitations due to high-dimensional, imbalanced, and non-linear data and can perform poorly in cases of large number predictive variables. To overcome these, we used a suite of classification tree models, a group of supervised machine learning models that can handle various types of data and handle interactions between predictive variables. Classification tree models like random forest, gradient boosting have been found useful in investigating the prevalence and associated risk factors of bovine viral diarrhea virus (Machado, Mendoza & Corbellini, 2015), swine pneumonia (Mollenhorst et al., 2019), and mastitis in dairy cattle (Hyde et al., 2020). In the study reported here, our objective was to explore the herd and cow level features associated with the multi-drug resistance and resistance phenotype shared between Salmonella, E. coli, or Enterococcus spp. using machine learning algorithms.

Materials & Methods

Farms surveys and sampling

The study was approved by the University of California, Davis’s Institutional Animal Care and Use Committee (protocol number 18019). Six dairy farms located in the San Joaquin Valley of California were reenrolled as a convenience sample and followed up for a second year after being part of an earlier study (Abu Aboud et al., 2016). Briefly, cull cows were identified for fecal sampling once during each season between 2015 and 2016, specifically during summer (July 1–September 30, 2015), fall (October 1–December 31, 2015), winter (January 1–March 31, 2016) and spring (April 1–June 30, 2016). The choice of sampling week to collect fecal samples from the cull cows during any of the four seasons was also by convenience. From the list of cows selected by the dairy farms for culling, 10 cows were randomly selected for fecal sampling on the day of their removal from the herd using a random number generator (Excel; Microsoft Corp., Redmond, WA, USA). Several lists of random numbers were prepared in multiples of 10 ranging from 11–20 to 91–100 cows and the respective list was selected depending on the number of cows presented for culling on the day of sampling. If a producer had less than 11 cows presented for culling, all the cows were sampled. An individual disposable sleeve was used to manually collect fecal samples from the rectum of each selected cow. The fecal samples were transported to the Aly Lab (Dairy Epi Lab) on wet ice for processing on the same day.

A survey was also completed with the help of the herd manager on the same sampling day. The survey questions were described in an earlier report (Pereira et al., 2019). Briefly, the questions targeted management of the herd in the previous 4 months and collected information on herd size, breed, rolling herd average, cull rate, frequency of culling per month, the proportion of cows sold for beef (compared to as dairy), proportion and reasons for condemnation of culled cows. The survey also collected information on the proportion of culled cows that received injectable medical treatments in the 3 weeks prior to culling, the role of dairy staff allowed to treat cows on the dairy, practices to avoid drug residue violations (use of specific drug types, following withdrawal times, milk and/or urine testing prior to the cow being culling, or other practices), tracking of drug withdrawal intervals, having a drug inventory system in place, and extra-label drug use (ELDU, frequency and familiarity). Finally, a backup of the herd record software was obtained to collect information on the culled cows’ milk production and health events. A relational database was used to house and merge data from the surveys, dairy records, and test results (Access; Microsoft Corp., Redmond, WA, USA).

Bacteriological culture

The California Animal Health and Food Safety (CAHFS) laboratory conducted all the study sample testing for Salmonella as described by Adaska et al. (2020). Briefly, 1 g of feces was inoculated into tetrathionate selective enrichment broth and incubated at 37 ± 2 °C. The next morning a cotton swab was used to inoculate the overnight broth onto XLD and XLT-4 plates and these were incubated overnight at 37 ± 2 °C. H2S positive, Salmonella suspect colonies from each set of plates were subcultured onto individual bi-plates (5% sheep blood agar-MacConkey agar) and incubated overnight at 37 ± 2 °C. One colony from each bi-plate was used for biochemical testing which included triple sugar iron, urea, motility indole ornithine, citrate, O-nitrophenyl-beta-D-galactopyranoside, and lysine iron agar slants. Serogrouping and serotyping were performed, using the White–Kauffmann–Le Minor scheme (Grimont & Weill, 2007) on colonies with biochemical test results consistent with Salmonella (Quinn et al., 2002).

Fecal samples were also cultured for E. coli and Enterococcus spp. isolation as described previously (Li et al., 2014). Briefly, a 40 mL solution of buffered peptone water containing 5 g of feces in a 50 mL polypropylene tube was homogenized using a mechanical shaker for 15 min before filtering using gauze. A total of 1000, 100 and 10 μl of the filtered solution were then streaked on CHROMAgar ECC (Chromagar, Paris) and Enterococcus Indoxyl-β-D-Glucoside agar (mEI) plates (Becton, Dickinson and Company, Franklin Lakes, NJ, USA) both incubated at 37 °C for 24 h. Reference strains ATCC 25922 (E. coli) and ATCC 29212 (Enterococcus faecalis) were plated on agar plates as positive controls. Two pure colonies were isolated of each species after presumptive colonies were confirmed using biochemical tests (E. coli were confirmed using urea, indole, triple sugar iron, Methyl Red–Voges–Proskauer and Simmons citrate; Enterococcus were confirmed using bile esculin, brain heart infusion agar, and growth in broth with and without 6.5% NaCl).

Antimicrobial susceptibility testing

Salmonella and E. coli bacterial resistance was evaluated with a broth microdilution method using a Gram-negative Sensititre plate (CMV2AGNF) and Enterococcus spp. evaluated on gram-positive Sensititre plate (CMV3AGPF) (Sensititre, Thermo Fisher Scientific, MA, USA) according to the manufacturer’s instructions and as described in previous studies (Li et al., 2018; Pereira et al., 2019). The minimum inhibitory concentration (MIC) values were the lowest concentrations of AMD that inhibited visible growth of bacteria. Interpretations of antimicrobial resistance were based on breakpoints recommended by the National Antimicrobial Resistance Monitoring System (https://www.cdc.gov/narms/antibiotics-tested.html; and https://www.ars.usda.gov/ARSUserFiles/60400520/NARMS/ABXEntero.pdf) and the Clinical and Laboratory Standards Institute (CLSI, 2014; CLSI, 2018). Due to the inherent resistance of Salmonella to cephalosporins, aminoglycosides, lincosamides, oxazolidinones and glycopeptides, drugs from these classes were excluded from the antimicrobial resistance analysis. In addition, the following drug classes to which E. coli is inherently resistant were excluded from the analysis: lincosamides, oxazolidinones, penicillins, streptogramins, glycopeptides. Similarly, the drug classes exclude due to the inherent resistance of Enterococcus spp. included cephalosporins, lincosamides, fluoroquinolones, aminoglycosides, aminocyclitols, sulfonamides, folate pathway antagonists. Isolates from any of the three species (Salmonella, E. coli, Enterococcus spp.) were identified as multi-drug resistant if resistance to at least one AMD in each of three or more drug classes was observed (Magiorakos et al., 2012).

Development of classification algorithms

Classification tree models were developed to test if herd management practices and features related to dairy cows can predict multi-drug resistance phenotype in Salmonella and fecal commensal E. coli, and Enterococcus spp. isolated from the same cows. A cow was considered to shed MDR bacteria if any of its Salmonella, Enterococcus spp. and/or E. coli isolates showed resistance for three or more antimicrobial classes (regardless of whether the species with resistance was Salmonella, Enterococcus spp. and/or E. coli), in which case the cow was labeled as ‘shedding bacteria with multi-drug resistance (MDR) phenotype’ (numerical label = 2). In contrast, if a cow shed bacteria that were resistant to only one or two antimicrobial classes (regardless of species) the bacteria isolated were labeled as ‘shedding bacteria with antimicrobial resistance (AMR) phenotype’ (numerical label = 1). If the bacteria isolated from a cow did not show any resistance across all three bacterial species to any AMD, the bacteria were labeled as non-resistant (numerical label = 0). The mutually exclusive definitions were necessary to develop a single model that predicts one of three resistance states MDR, AMR, or no resistance. Similarly, cows were also classified based on resistance phenotypes separately observed in each bacterial species isolated (three separate classification labels based on resistance phenotypes of Salmonella, Enterococcus spp. and E. coli isolates). Finally, classification labels were also generated based on resistant phenotypes observed in either commensal bacteria (Enterococcus spp. and E. coli) shed in feces collectively. Using features from herd surveys, classification tree models were trained to predict the MDR phenotype of bacteria shed in the feces of the study cows.

Three machine learning algorithms, specifically decision tree classifier (DTC), random forest (RF) and gradient boosting (GB) were developed to explore the risk factors for resistance phenotypes in the study isolates using herd survey data, specifically to predict the multilabel outcome based on the resistance phenotype of bacteria shed by cows (non-resistant, AMR, and MDR). For each algorithm, data from the entire study cohort described by Pereira et al. (238 cows) were used, except models specific to predicting AMR phenotypes in Salmonella were restricted to the cohort of 58 Salmonella positive cows. Table 1 describes all classification algorithms trained and developed for various bacteria-specific outcomes.

Table 1 Hypertuning of model parameters and validation.

Model parameters	Output definition based on resistance for number of antimicrobial classes	Salmonella
(n = 58)	E. coli
(n = 238)	Enterococcus spp.
(n = 238)	Combined resistance in E. coli and Enterococcus spp. (n = 238)	Resistance in antimicrobial classes in either Salmonella or E. coli or Enterococcus spp.
(n = 238)	
Parameter explanation	Parameter values tested	Best performing model parameters	
Decision tree classifier	
Criterion	The function to measure the quality of a split	gini, entropy	gini	entropy	entropy	gini	gini	
Splitter	The strategy used to choose the split at each node.	best, random	best	best	best	random	random	
Maximum depth	The maximum depth of the tree	10, 20, 30, 40, 45, 50, 70	10	45	50	45	10	
Minimum split	The minimum number of samples required to split an internal node	2, 3, 4, 6	6	6	6	3	3	
Maximum features	The number of features to consider when looking for the best split	auto, sqrt	auto	auto	auto	sqrt	sqrt	
Minimum leaf	The minimum number of samples required to be at a leaf node	1, 3, 4, 6, 7, 8	4	4	6	1	7	
Model performance for holdout dataset	
n	Number of samples in holdout dataset		19	73	73	73	73	
Precision	Positive predictive value		0.72	0.65	0.51	0.44	0.47	
Recall	Sensitivity		0.79	0.68	0.52	0.44	0.47	
F1-score	Harmonic mean of PPV and sensitivity		0.75	0.66	0.50	0.44	0.47	
Random forest	
Bootstrap	Whether bootstrap samples are used when building trees	True, False	FALSE	TRUE	TRUE	TRUE	TRUE	
Criterion	The function to measure the quality of a split	gini, entropy	gini	entropy	entropy	entropy	entropy	
Maximum depth	The maximum depth of the tree	10, 20, 30, 40, 45, 50, 70	30	50	30	40	10	
Maximum features	The number of features to consider when looking for the best split	auto, sqrt	sqrt	sqrt	sqrt	auto	auto	
Minimum leaf	The minimum number of samples required to be at a leaf node	1, 3, 4, 6, 7, 8	1	8	4	8	8	
Minimum split	The minimum number of samples required to split an internal node	2, 3, 4, 6	3	3	2	4	4	
Number of estimators	The number of trees in the forest	100, 200, 300, 500	200	100	100	100	100	
Model performance for holdout dataset	
n	Number of samples in holdout dataset		19	73	73	73	73	
Precision	Positive predictive value		0.66	0.62	0.46	0.47	0.41	
Recall	Sensitivity		0.74	0.67	0.47	0.48	0.41	
F1-score	Harmonic mean of PPV and sensitivity		0.69	0.61	0.46	0.47	0.38	
Gradient boosting (XGBoost framework)	
Column sample	Subsample ratio of columns when constructing each tree	0.2, 0.1,0.15, 0.4, 0.7	0.1	0.2	0.2	0.4	0.1	
Gamma	Minimum loss reduction required to make a further partition on a leaf node of the tree	0.0, 0.1, 0.2, 0.4, 0.45, 0.5, 0.6, 0.7	0.1	0.1	0.2	0.6	0.5	
Learning rate	Boosting learning rate	0.001, 0.002, 0.005, 0.008, 0.01, 0.02, 0.05, 0.10, 0.25, 0.5	0.25	0.5	0.008	0.5	0.05	
Maximum depth	Maximum tree depth for base learners	3, 4, 7, 8, 9, 10, 15, 20	3	8	8	7	7	
Minimum child weight	Minimum sum of instance weight (hessian) needed in a child	1, 3, 5, 7	3	1	1	1	7	
Number of estimators	Number of boosting rounds	3, 5, 10, 30, 40, 50, 100	5	30	10	3	30	
Objective	Learning task, binary, multiple, etc.	multi:softprob	multi:softprob	multi:softprob	multi:softprob	multi:softprob	multi:softprob	
Model performance for holdout dataset	
n	Number of samples in holdout dataset		19	73	73	73	73	
Precision	Positive predictive value		0.61	0.59	0.51	0.49	0.46	
Recall	Sensitivity		0.68	0.6	0.51	0.48	0.44	
F1-score	Harmonic mean of PPV and sensitivity		0.65	0.59	0.51	0.48	0.38	
Note:

Classification algorithms trained and tested to predict multidrug resistance phenotypes from bacterial isolates for bacterial species and groups of bacteria. Parenthesis (n) sample size of the number of cows for each model. For each bacterial and bacterial group model, hyper-tuning of the decision tree classifier, random forest, and XGBoost models is presented. The table shows parameters tuned, values tested for tuning models, best model parameters, and the performance of the selected model in terms of precision, recall and F1-score for the holdout dataset.

For all three classification algorithms, 25 features related to herd management and 12 features related to individual cows collected from the survey were used as predictive features (Table 2). The DTC generates an optimum tree based on attributes by recursive selections to split data into classes and it was only used to visualize the optimum tree and as a contrast to the remaining classification tree models (RF and GB) that prevent overfitting, unlike DTC. The RF and GB algorithms both generate a series of recursive trees of binary splits for randomly sampled predictor variables. While all tree classification algorithms handle interaction effects between predictors, within GB, boosting builds and combines collective models improving the predictive performance of many weak models substantially, and fits complex nonlinear relationships (Elith, Leathwick & Hastie, 2008). For the validation of each algorithm, the data was split into training and validation datasets. To identify the best hyperparameters of the classification algorithms (hyper-tuning), a grid search was implemented on the training dataset. Grid search is a tuning technique that computes optimum values of model parameters automatically by an exhaustive search performed on a set of parameter values. Training datasets, composed of 80% of the data, were randomly selected for gird search, maintaining the outcome proportional to the original dataset, with three-fold cross-validation (internal validation). Model parameters tested for hyper-tuning of DTC, RF, and GB are given in Table 1. The best performing model parameters were chosen based on the accuracy of the model. For each algorithm, the external validation of the best performing model was done on the validation dataset (20% remaining random sample of the original dataset) to quantify the performance of the model on the completely independent validation dataset. The DTC and RF models were implemented using the Scikit-learn machine learning package (Pedregosa et al., 2011) and GB was implanted using XGBoost in python (Chen & Guestrin, 2016).

Table 2 Predictive features.

Cow related features	Herd related features	
Low milk production cull (LowMilkCull)	Milking herd size (HerdSize)	
Reproduction cull (ReproCull)	Milk production level (RollingHerdAvg)	
Lameness cull (LameCull)	Holstein Breed (Holstein)	
Mastitis cull (MastitisCull)	Jersey Breed (Jersey)	
Other reasons cull (OtherCull)	Percent culled monthly (CullPctMonth)	
Antimicrobial Drug Use for cull condition (AMD)	Times culled monthly (CullTimesMonth)	
Anti-inflammatory treatment for condition (Ani-Inf)	Main cull reason disease	
No-Treatment for condition (No-Treatment)	Percent culled sold for beef (PctCullBeef)	
Other treatment for condition (Other)	Percent carcasses condemned (PctCullCondemned)	
	Percent culled injected within 2~3 weeks (PctInject)	
	Veterinarian gives sick cow treatments (VetTreats)	
	Dairy manager gives sick cow treatments (ManagerTreats)	
	Staff gives sick cow treatments (StaffTreats)	
	Prevent Residue by avoiding specific drugs (ResiduePrevent)	
	Chalk on cows to track drug withdrawal (Chalk4Withdrawal)	
	Keep drug inventory (Inventory)	
	Penicillin	
	Ceftiofur	
	Tetracycline	
	Antibiotics used separately (SeparateUse)	
	Antibiotics combinations used (CombinationUse)	
	Track antibiotic dose used (TrackAntibioticDose)	
	Track antibiotic route used (TrackAntibioticRoute)	
	Familiarity with ELDU (FamiliarELDU)	
	Frequency of ELDU (FreqELDU)	
	No ELDU (NoELDU)	
	Number of cull cows culled today (NumberCulled)	
	Use of Salmonella vaccine (SalmonellaVaccine)	
	Sampling Season	
Note:

Dairy cattle and herd related features used as predictors in classification models.

Validated models were eventually fit on the complete dataset to produce the final predictions. The relative influence (importance) of features for the random forest model was estimated using average gini importance, permutation, and feature drop methods. The gini importance for a feature is defined as the sum over the number of splits (across all tress) that include the feature, proportionally to the number of samples it splits (Breiman, 2001). The gradient boosting model with the XGboost platform was evaluated using Shapely Additive Explanations (SHAP) that assigned each predictive feature an additive feature unifying six existing methods (Lundberg & Lee, 2017). Partial dependence of gradient boosted model prediction on model features (expected output response trend as a function of feature) was explored to understand the associations of the herd and cow-related features with predictions (Friedman, 2001). The Python code used for pre-processing the data, training and validating models and generating figures can be found here in the Zenodo repository: DOI 10.5281/zenodo.4387017.

Results

From the six dairy herds, Salmonella isolates were detected in the feces of 58 cows (24.4% ± 2.8 out of 238 cows). Two herds had no Salmonella positive samples throughout the study period while for others the prevalence ranged from 12.5% (±5.2, n = 40) to 70.0% (±7.2, n = 40).

The most common reason reported for culling was low milk production (65.1%, ±3.1) followed by poor reproduction (31.1%, ±3.0). Lameness (10.5%, ±1.9) and mastitis (10.1%, ±1.9) were the other reasons reported in the survey. Of the sampled cull cows, 15.5% (±2.3) were reported as having received AMD as part of a treatment protocol for the condition resulting in their culling decision. In contrast, 5.9% (±1.5) received anti-inflammatory drugs for the condition resulting in their culling decision.

Distribution of antimicrobial resistance

Predominant resistance phenotypes detected

The prevalence of AMR and MDR phenotypes in Salmonella was reported previously by Pereira et al. (2019). Briefly, tetracyclines were the most prevalent drug class for which Salmonella were resistant and 12% of the study isolates tested positive for MDR Salmonella (Pereira et al., 2019). Within Enterococcus spp. isolates, the most common AMR phenotype detected was for nitrofuran antimicrobials, (10.83%, ± 2.47). The most common type of MDR phenotype in Enterococcus spp. isolates was resistance to oxazolidinones, nitrofuran antibacterials, and macrolides (5.73%, ±1.85). Frequencies of all resistance phenotypes observed in Enterococcus spp. are presented in Table 3.

Table 3 Resistance phenotypes detected in Enterococcus spp. isolates.

Resistance phenotypes observed in Enterococcus spp. (Antimicrobial class)	Number of cows (total N = 157)	The proportion of cows (%)	95% CI	
Nitrofuran antibacterial	17	10.83	[5.967–15.689]	
Macrolides	15	9.55	[4.956–14.152]	
Nitrofuran antibacterial, Macrolides	15	9.55	[4.956–14.152]	
Oxazolidinones, Nitrofuran antibacterial, Macrolides†	9	5.73	[2.096–9.369]	
Oxazolidinones, Nitrofuran antibacterial	8	5.1	[1.656–8.535]	
Tetracyclines	7	4.46	[1.23–7.687]	
Oxazolidinones	6	3.82	[0.823–6.821]	
Tetracyclines, Nitrofuran antibacterial	6	3.82	[0.823–6.821]	
Tetracyclines, Nitrofuran antibacterial, Macrolides†	5	3.18	[0.438–5.931]	
Streptogramin, Oxazolidinones, Nitrofuran antibacterial, Macrolides†	4	2.55	[0.083–5.013]	
Tetracyclines, Amphenicols, Oxazolidinones, Nitrofuran antibacterial†	4	2.55	[0.083–5.013]	
Streptogramin, Nitrofuran antibacterial	4	2.55	[0.083–5.013]	
Tetracyclines, Amphenicols, Nitrofuran antibacterial, Macrolides†	3	1.91	[0.0–4.052]	
Amphenicols, Oxazolidinones, Nitrofuran antibacterial, Macrolides†	3	1.91	[0.0–4.052]	
Tetracyclines, Macrolides	3	1.91	[0.0–4.052]	
Amphenicols, Oxazolidinones, Nitrofuran antibacterial†	3	1.91	[0.0–4.052]	
Oxazolidinones, Macrolides	3	1.91	[0.0–4.052]	
Amphenicols, Streptogramin, Oxazolidinones, Nitrofuran antibacterial, Macrolides†	3	1.91	[0.0–4.052]	
Streptogramin	2	1.27	[0.0–3.028]	
Tetracyclines, Oxazolidinones	2	1.27	[0.0–3.028]	
Tetracyclines, Amphenicols, Nitrofuran antibacterial†	2	1.27	[0.0–3.028]	
Tetracyclines, Amphenicols, Macrolides, Streptogramin, Oxazolidinones, Nitrofuran antibacterial, Macrolides†	2	1.27	[0.0–3.028]	
Amphenicols, Streptogramin, Oxazolidinones, Nitrofuran antibacterial†	2	1.27	[0.0–3.028]	
Tetracyclines,Oxazolidinones, Nitrofuran antibacterial, Macrolides†	2	1.27	[0.0–3.028]	
Other single isolates of MDR phenotypes**	23	14.72	[9.118–20.180]	
Other single isolates of AMR phenotypes*	4	2.55	[0.083–5.013]	
Notes:

* Other single isolates of amr phenotypes from Enterococcus sp. isolates: (1) amphenicols, nitrofuran antibacterial (2) streptogramin, oxazolidinones (3) amphenicols, macrolides (4) macrolides, oxazolidinone.

** Other single isolates of MDR phenotypes from Enterococcus spp. isolates: (1) tetracyclines, amphenicols, oxazolidinones, nitrofuran antibacterial, macrolides (2) oxazolidinones, nitrofuran antibacterial, macrolides, glycopeptides (3) tetracyclines, amphenicols, oxazolidinones, macrolides (4) streptogramin, oxazolidinones, nitrofuran antibacterial (5) tetracyclines, streptogramin, nitrofuran antibacterial, macrolides (6) tetracyclines, amphenicols, streptogramin, oxazolidinones, nitrofuran antibacterial, macrolides (7) tetracyclines, amphenicols, macrolides, streptogramin, oxazolidinones, macrolides (8) amphenicols, macrolides, streptogramin, oxazolidinones, nitrofuran antibacterial (9) macrolides, streptogramin, oxazolidinones, nitrofuran antibacterials, macrolides (10) amphenicols, nitrofuran antibacterial, macrolides (11) oxazolidinones, nitrofuran antibacterial, glycopeptides (12) amphenicols, oxazolidinones, macrolides (13) tetracyclines, macrolides, oxazolidinones, nitrofuran antibacterial, macrolides (14) tetracyclines, streptogramin, nitrofuran antibacterial (15) tetracyclines, macrolides, nitrofuran antibacterial, macrolides (16) amphenicols, streptogramin, macrolides (17) tetracyclines, streptogramin, oxazolidinones (18) streptogramin, nitrofuran antibacterial, macrolides (19) tetracyclines, macrolides, oxazolidinones, nitrofuran antibacterial (20) tetracyclines, oxazolidinones, nitrofuran antibacterial (21) tetracyclines, amphenicols, macrolides, oxazolidinones, nitrofuran antibacterial (22) tetracyclines, amphenicols, macrolides, nitrofuran antibacterial (23) amphenicols, macrolides, nitrofuran antibacterial.

† Phenotypes that are multi-drug resistant.

Tetracyclines resistance was the most prevalent phenotype in E. coli isolates (30.86%, ±5.13). E. coli isolates from seven cows (8.64%, ±3.12), showed MDR phenotypes, and only one MDR phenotype was detected more than once (aminoglycosides, tetracyclines, amphenicols, 2.47%, ±1.72). Frequencies of all resistance phenotypes observed in E. coli are presented in Table 4.

Table 4 Resistant phenotypes detected in E. coli isolates.

Resistant phenotypes observed in E. coli (Antimicrobial class)	Number of cows (total N = 81)	The proportion of cows (%)	95% CI	
Tetracyclines	25	30.86	[20.805–40.924]	
Aminoglycosides	11	13.58	[6.12–21.041]	
Cephalosporins	9	11.11	[4.267–17.955]	
Aminoglycosides, Tetracyclines	7	8.64	[2.523–14.761]	
Folate pathway antagonist	5	6.17	[0.932–11.414]	
Amphenicols	5	6.17	[0.932–11.414]	
Tetracyclines, Cephalosporins	2	2.47	[0.0–5.849]	
Aminoglycosides, Tetracyclines, Amphenicols†	2	2.47	[0.0–5.849]	
Other single isolates of AMR phenotypes*	5	6.17	[0.932–11.374]	
Other single isolates of MDR phenotypes**	5	6.17	[0.932–11.414]	
Notes:

* Other single isolates of AMR phenotypes from E. coli: (1) quinolones, aminoglycosides (2) amphenicols, tetracyclines (3) tetracyclines, folate pathway antagonist (6) cephalosporins, fluoroquinolones (7) amphenicols, folate pathway antagonist.

** Other single isolates of mdr phenotypes from E. coli: (1) amphenicols, folate pathway antagonist, aminoglycosides (2) amphenicols, tetracyclines, cephalosporins, aminoglycosides (3) amphenicols, tetracyclines, folate pathway antagonist (4) amphenicols, tetracyclines, folate pathway antagonist, aminoglycosides (5) tetracyclines, cephalosporins, fluoroquinolones, quinolones, aminoglycosides.

† Phenotypes that are multi-drug resistant.

Seasonality of multi-drug resistance

The overall prevalence of shedding MDR bacteria (Salmonella, E. coli, and Enterococcus spp.) in cows was 30.54%, (±2.97, n = 238). The prevalence of cows shedding bacteria that are resistant to <=2 antimicrobial drug classes (AMR) was 43.93% (±3.21, n = 238). The highest prevalence of MDR was seen in the summer (50.00% ± 11.18) followed by fall (34.00%, ±4.73). The highest prevalence of shedding MDR bacteria was seen in herd 4 (52.00%, ±7.89) and the lowest was seen in herd 3 (10.00% ± 4.74). Herds 1 and 2 each showed the lowest prevalence of shedding AMR bacteria (37.50% ± 7.65), while herd 4 showed the highest prevalence of shedding AMR (50.00%, ±7.90). The seasonal prevalence of MDR and AMR resistance across three bacterial species is presented in Fig. S1.

The annual prevalence of cows shedding MDR Salmonella was 8.62% (±3.65, n = 58). The highest prevalence for MDR was detected in the winter season (10.53%, ±7.04, n = 19) and the highest AMR prevalence was detected in fall (42.31% ± 9.68, n = 26). Salmonella isolated from cows in the summer season did not show any AMR or MDR phenotypes (Fig. 1A). The annual prevalence for MDR phenotypes within E. coli isolates was 2.92% (±1.09, n = 238). The highest prevalence for MDR phenotypes in E. coli was detected in winter (4.08% ± 2.82, n = 49) and the highest AMR prevalence in E. coli was detected in spring (34.28% ± 5.67, n = 70, Fig. 1B). The annual prevalence for MDR phenotypes within Enterococcus spp. isolates was 26.35% (±2.84, n = 238). The highest prevalence for MDR Enterococcus spp. was detected in the summer season (50.00% ± 11.18, n = 20) and the highest AMR prevalence in Enterococcus spp. was detected in the winter season (48.97% ± 7.14, n = 49, Fig. 1C).

Figure 1 Seasonal variation in the prevalence of multidrug antimicrobial resistance.

Seasonal variation in the prevalence of multidrug antimicrobial resistance (MDR; resistance to three or more drug classes), and antimicrobial resistance (AMR; resistance to one or two drug classes only) in Salmonella (A), E. coli (B) and Enterococcus spp. (C) isolates from six California dairy herds. Orange and green dashed lines show the annual average prevalence of MDR and AMR in all six herds respectively. The proportion of cows that did not show any resistance are the inverse of the sum of MDR and AMR proportions and not shown in the figure. Point estimates and single standard error deviation are represented by circles and whiskers respectively.

Antimicrobial resistance phenotype shared between bacterial isolates

Study found no perfectly shared resistance phenotype between the study isolates. Within the commensal bacteria (E. coli or Enterococcus spp.), resistance to tetracycline was the most prevalent (11 cows), while shared resistance to each of the drugs kanamycin, streptomycin, and chloramphenicol was observed once. However, six individual cows were detected sharing tetracycline resistance phenotypes between Salmonella and one or both commensal bacteria (E. coli or Enterococcus spp.). Of the six cows, five were from a single herd (herd 4) and three shed MDR bacteria (Table 5). However, Salmonella, and one or both commensal species, were found to share resistance to only one drug, tetracycline, in six cows across the study. Of the six cows, only three had bacterial species with MDR phenotypes detected and in two of these cows the MDR profile included tetracycline.

Table 5 Tetracycline antimicrobial resistance phenotype shared (highlighted in grey) between commensal bacteria (Enterococcus spp., E. coli) and Salmonella.

Resistance Phenotypes observed	
Salmonella	Enterococcus spp.	E. coli	Herd ID	Resistance	
Tetracyclines	Nitrofuran antibacterial	Aminoglycosides, Tetracyclines	4	AMR	
Tetracyclines	Macrolides, Oxazolidinones, Nitrofuran antibacterial†	Tetracyclines, Amphenicols	4	MDR	
Tetracyclines, Penicillins	Macrolides, Nitrofuran antibacterial	Aminoglycosides, Tetracyclines, Folate pathway antagonist, Amphenicols†	4	MDR	
Tetracyclines, Folate pathway antagonist	Macrolides, Amphenicols	Tetracyclines	4	AMR	
Tetracyclines	Tetracyclines, Nitrofuran antibacterial	Tetracyclines, Folate pathway antagonist, Amphenicols†	4	MDR	
Tetracyclines	Streptogramin, Oxazolidinones	Tetracyclines	6	AMR	
Notes:

† Phenotypes that are multi-drug phenotypes.

Each row represents bacterial resistance phenotypes of bacterial isolates from a single culled dairy cow.

Model performance and tuning results

Grid search of model parameters with 3-fold cross-validation yielded satisfactory results in classifying MDR cows based on herd management practices and cow-related features. The sensitivity of models ranged from 0.47 to 0.74 with precision ranging from 0.46 to 0.66. Details related to hyperparameters, best performing decision tree classifiers, random forest, and XGboost models and the performance of the selected models are given in Table 1.

Association between herd management practices and cow-related features with shedding multi-drug resistant bacteria

Decision tree classification models

For all DTC models, the impurity (gini), a measure of the heterogeneity of the outcome in a subset of samples resulting from a split in a decision tree, was reduced in samples the most by the rolling herd average milk production, denoting its highest position in decision trees generated by all five models (Fig. 1, Figs. S2–S5). Other management features that formed nodes showing high measures of split quality (gini) including the proportion of Jersey cows in the herd, administration of tetracyclines, monthly culling percentage, culling frequency, and the number of culled individuals. Of the five DCT models, all but the model for overall resistance across Salmonella and commensal bacterial species were able to generate nodes that classify cows based on their fecal shedding of bacteria to a single class of AMD, or not resistant (Figs. S2–S5) with 100% purity (homogenous subsets). While only the DCT model for commensal bacteria, resulted in pure nodes for MDR cows, none of the other models were able to classify cows into pure MDR nodes (Fig. S2). Figure 2 shows the optimum decision tree for classifying cows into MDR, AMR, or not resistant based on phenotypes of all bacteria (Salmonella, Enterococcus spp., E. coli).

Figure 2 Optimum decision tree to classify cows shedding multi-drug resistant (MDR), antimicrobial-resistant (AMR), and non-resistant Salmonella, Enterococcus spp., E. coli based on management practices observed in Californian dairy herds.

Nodes are represented by stacked histograms depicting distribution samples in the data (AMR, MDR, no resistance), followed by optimum decision point (pointer on the x-axis). Arrows on the left and right indicate lesser and greater than the decision point respectively. Final nodes are represented by pie chart with distribution of samples. Factor acronym definitions are described in Table 2.

Random forest models

Results of random forest models indicated common herd management practices that influence the shedding of MDR and AMR phenotypes of Salmonella, Enterococcus spp., and E. coli collectively, as well as individually (Fig. 3). The number of culled animals, monthly culling frequency and percentage, herd size, and proportion of Holstein cows in the herd were found to be influential herd characteristics (top ten features by relative influence) predicting MDR phenotypes in all algorithms. Random forest algorithms for predicting AMR phenotype in Enterococcus spp., commensal bacterial species combined, and in all bacterial species showed the same top ten most influential herd management features. Individual-level features such as culling due to milk production, mastitis, reproductive and other reasons appeared important for predicting resistance phenotypes in Salmonella. The use of the chalk method for withdrawal determination was in the top ten most influential features and for the E. coli and Salmonella models.

Figure 3 Top ten herd management practices based on variable importance (Gini coefficient) in classifying cows shedding multi-drug resistant (MDR), antimicrobial-resistant (AMR), and non-resistant for Salmonella, Enterococcus spp., E. coli in Cal.

Factor acronym definitions described in Table 2.

Gradient boosting classification models

Herd characteristics that showed higher variable importance in SHAP values for predicting cows shedding MDR resistance bacteria based on all bacterial species were herd size, the proportion of Jersey cows, sampling season, the frequency of extra-label drug use, rolling herd average, and culling related features. These features also showed high marginal contributions in predicting AMR phenotypes. For predicting AMR phenotypes, features such as the number of monthly veterinary treatments, antibiotic dose tracking, and the proportion of Holstein cows also showed higher SHAP values (Fig. 4).

Figure 4 Mean SHAP values depicting the impact of herd management practices on predicting multi-drug resistant phenotype in either Salmonella, Enterococcus spp., and E. coli shed in dairy cows for Gradient boosting classification (XGboost) model.

Factor acronym definitions described in Table 2.

Partial dependence of gradient boosting model prediction on important features showed the possible relationships of these herd and cow-related features in predicting the AMR of their fecal bacteria as MDR. Higher culling frequency and monthly culling percentages were associated with cows with MDR phenotypes from all bacteria, whereas an overall increase in the number of culled animals from the herd showed a negative trend with classifying a cow as shedding MDR bacteria. Winter season was negatively associated with MDR phenotype bacteria shed by cows compared to cows sampled in Spring. Similarly, herds with more than 10,432 kgs of rolling average milk production showed a positive trend with MDR positive cows. Cows from herds with a higher proportion of Jersey cows were negatively associated with being classified as shedding MDR bacteria by the gradient boosting algorithm (Fig. 5). Within other herd characteristics that were identified as important, herd size showed a varying trend with classifying cows shedding MDR bacteria, with some herd sizes showing a positive association of classifying cows as shedding MDR bacteria (Fig. 5).

Figure 5 Partial dependence indicating the association of top six predictive herd management practices in classifying cows as multi-drug resistant phenotype in either Salmonella, Enterococcus spp. and E. coli shed in dairy cows for Gradient boost.

Partial dependence plots are generated for values presented in the data resulting in the non-linear x-axis. Blue shaded region and error bars represents standard deviation of partial dependence (n = 238).

For predicting MDR phenotypes in Salmonella shed by cows, rolling herd average milk production, sampling season, chalk methods for tracking withdrawal, monthly culling frequency, and Salmonella vaccine were found to be important predictive features with high SHAP values (Fig. 6). The exploration into partial dependence of these features gave insights into relationships of feature values with classifying a cow as shedding MDR phenotype Salmonella (Fig. 7). Increasing rolling herd average milk production and monthly culling percentage was positively associated with MDR phenotypic Salmonella in cow feces. Similarly, cows sampled in spring were more likely to be classified as shedding MDR Salmonella.

Figure 6 Mean SHAP values depicting the impact of herd management practices on predicting multi-drug resistant phenotype in Salmonella shed in dairy cows for Gradient boosting classification (XGboost) model.

Factor acronym definitions described in Table 2.

Figure 7 Partial dependence indicating the association of top-six predictive herd management practices in classifying cows as multi-drug resistant phenotype in Salmonella shed in dairy cows for Gradient boosting classification (XGboost) model.

Partial dependence plots are generated for values presented in the data resulting in the non-linear x-axis. Blue shaded region and error bars represent the standard deviation of partial dependence (n = 58).

Models predicting phenotypes in commensals E. coli, and Enterococcus spp.; the number of culled animals in the previous year was always in the top ten most important features to classify cows as shedding MDR bacteria (Figs. S5–S8). Rolling herd average milk production features describing culling practices, and herd size, consistently featured as important in classifying cows as shedding MDR phenotypic bacteria for all the other three models (Commensals, E. coli and Enterococcus spp.). The frequency of extra-label drug use was an important feature for models separately predicting MDR in Enterococcus spp. (Fig. S7) and E. coli shed by cows.

Discussion

The current study investigated antimicrobial resistance phenotypes between bacteria shed in dairy cattle using decision tree classification algorithms. A simple decision tree model does tend to find the best fit for the training data, but the splitting process rarely is generalizable to other data. A random forest model, which is bagging of decision trees, and a boosting classification model, which is boosting decision trees, tend to perform better on testing data and can help us identify the generalizable conclusion. In this analysis we explored these models step by step from a simple decision tree to generalized boosting trees to find important management-related factors that might affect the distribution of multidrug resistance in dairy cattle herds.

A unique aspect of the current study is use of the aforementioned algorithms to distinguish between the resistance profiles (no resistance, antimicrobial resistance and multidrug resistance) of a pathogenic bacteria, Salmonella, and commensal bacteria (E. coli and Enterococcus spp.) isolates from feces of cull dairy cows. Previous investigations were restricted to understanding herd and cow level characteristics with the resistant phenotypes in Salmonella shed by cull dairy cows (Pereira et al., 2019). We explored weather considering the resistance phenotypes for these three bacterial species together, and separately, to identify associations between herd management practices and prevalence of resistance phenotypes. Although, none of the isolates had shared phenotype resistance, the six cows that had tetracycline resistance in at least two of the three bacteria studied should be explored further with whole genome sequencing and follow up studies that employ metagenomic analyses on the microbiome.

The three machine learning algorithms tested in this study indicated that the overall distributions of three resistance phenotypes classified in this study as MDR, AMR, or no resistance were mainly governed by resistance in Enterococcus spp., which showed the highest prevalence of MDR and AMR phenotypes compared to Salmonella and E. coli. The latter may be explained by Enterococcus spp. that are known to have frequent MDR phenotypes such as E. faecium. Comparative feature importance plots for all random forest models developed for these bacterial groups indicated the same, where similar features are found to be important for the model predicting MDR in all bacteria together, in commensals together, and in Enterococcus spp..

Herd size has been already identified as associated with higher odds of detecting Salmonella resistant to tetracycline (Pereira et al., 2019). In the current study, we showed that herd size was also positively associated with detecting MDR phenotypes in Salmonella as well as the commensal bacteria E. coli and Enterococcus spp. Salmonellosis is known to be associated with poor milk production and reproduction and hence increased risk for diseases, such as mastitis and infertility, and AMD treatments which may explain the 12% MDR in Salmonella isolates from the study cows (Lanzas et al., 2010). Similarly, Salmonella has been associated with clinical disease in both adult and young dairy cattle and beef cows (Divers & Peek, 2007; Pender, 2003; Roy et al., 2001; Smith, 2014). Percent cull and rate would reflect the increase in culling due to diseases better than actual numbers culled. This is evident from the importance of all three in the random forest model for Salmonella MDR phenotypes. In contrast, the total number culled was negatively associated with shedding multidrug resistant bacteria which may reflect producer decisions to prioritize the culling of otherwise healthy but low-producing cows based on milk or beef market value (with respect to dairy beef), amongst other factors. Other than market dynamics including milk or beef demand, underlying herd health or management reasons may explain the opposing trends of both percent and number culled, and the outcome MDR in cull cow fecal bacteria. Herds with MDR in their cull cow fecal bacteria hence need to be explored further to identify mechanisms that eventually increase culling rates. It is worthy to note that the random forest algorithm identified diseases of relative importance for MDR in Salmonella but not commensals. However, caution should be exercised with interpreting findings from this specific analysis due to the inability to ascertain that such diseases preceded the Salmonella shedding and specifically MDR status. The combined random forest model for all 3 species however did not show diseases as correlated with MDR, which may be due to the overall effect of the commensals in the dataset.

SHAP values ranked variables by importance for classifying resistance type (either MDR, AMR, or no resistance). However, in the case of rolling herd average milk production (RHA), SHAP values were allocated only for predicting MDR or AMR in models for Salmonella and Enterococcus and commensals, showing zero importance in predicting absence of any resistance. The RHA is hence more important in terms of identifying any resistance type (MDR or AMR) versus no resistance. All three algorithms (DCT, RF and XGboost) indicated a high importance of RHA in predicting MDR in fecal bacteria of cull cows. RHA is a dairy performance indicator affected by multiple herd characteristics such as age and breed structure. Studies have indicated stress related to higher production which may increase the chance of certain health conditions subsequently increasing the risk of antimicrobial drug use and hence bacterial resistance (Robbins et al., 2016). In addition, the association between RHA and MDR presented by all the models here is conditioned upon other features that follow the splits further down the classification tree. In contrast, for E. coli, RHA was important in identifying all resistance classifications (AMR, MDR, no resistance).

XGBoost results show that season was correlated with MDR in all 3 species. Specifically, Spring and Fall had a greater correlation with MDR compared to Summer and Winter, with Winter being the least correlated. Similar findings have been observed with a risk of disease in calves in Spring and Fall with bovine respiratory disease (Cummings et al., 2019; Dubrovsky et al., 2019; Maier et al., 2019). The current study’s bacterial species, specifically Salmonella, E. coli and Enterococcus spp., shared no specific MDR profiles; however, shared tetracycline resistance was detected. Further studies employing whole genome sequencing and metagenomics on the microbiome may explore factors that explain such shared resistance.

Conclusions

The current study characterized dairy cattle herd management practices that were associated with fecal shedding of multi-drug resistant bacteria. We identified generalizable and distant associations between pathogenic Salmonella and commensal bacteria. The data-driven suite of machine learning algorithms used here can help develop data-informed tools for better decision making, and risk assessment related to antibacterial resistant shedding by cows.

Supplemental Information

Supplemental Information 1 Supplementary Figures.

Click here for additional data file.

The authors acknowledge the study dairies’ owners, herd managers, and staff from California dairies participating in the study. Any opinions, findings, conclusions, or recommendations expressed in this publication are those of the author(s) and do not necessarily reflect the view of the U.S. Department of Agriculture.

Additional Information and Declarations

Competing Interests

Author Contributions

Animal Ethics

Data Availability

Sharif S. Aly is an Academic Editor for PeerJ.

Pranav S. Pandit conceived and designed the experiments, performed the experiments, analyzed the data, prepared figures and/or tables, authored or reviewed drafts of the paper, and approved the final draft.

Deniece R. Williams conceived and designed the experiments, authored or reviewed drafts of the paper, and approved the final draft.

Paul Rossitto conceived and designed the experiments, authored or reviewed drafts of the paper, and approved the final draft.

John M. Adaska conceived and designed the experiments, authored or reviewed drafts of the paper, and approved the final draft.

Richard Pereira conceived and designed the experiments, authored or reviewed drafts of the paper, and approved the final draft.

Terry W. Lehenbauer conceived and designed the experiments, authored or reviewed drafts of the paper, and approved the final draft.

Barbara A. Byrne conceived and designed the experiments, authored or reviewed drafts of the paper, and approved the final draft.

Xunde Li conceived and designed the experiments, authored or reviewed drafts of the paper, and approved the final draft.

Edward R. Atwill conceived and designed the experiments, authored or reviewed drafts of the paper, and approved the final draft.

Sharif S. Aly conceived and designed the experiments, performed the experiments, analyzed the data, authored or reviewed drafts of the paper, and approved the final draft.

The following information was supplied relating to ethical approvals (i.e., approving body and any reference numbers):

The study was approved by the University of California, Davis’s Institutional Animal Care and Use Committee (protocol number 18019).

The following information was supplied regarding data availability:

The de-identified data was only shared for peer review as the dairy owners did not consent to publish it alongside the article.

The code used for the development of models, generating results, tables, and figures are available at Zenodo: Pranav Pandit. (2020). PanditPranav/MultiDrugResistance_DairyCattle: First release for submission of the manuscript (Version V.1.0). Zenodo. DOI 10.5281/zenodo.4387016.

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
