# Peer review of "Dairy management practices associated with multi-drug resistant fecal commensals and Salmonella in cull cows: a machine learning approach"

_PeerJ, doi:10.7717/peerj.11732_

## Round 0.1 · original submission · Major Revisions

Dear Dr. Pandit and colleagues:

Thanks for submitting your manuscript to PeerJ. I have now received three independent reviews of your work, and as you will see, the reviewers raised some concerns about the research. Despite this, these reviewers are quite optimistic about your work and the potential impact it will have on research studying herd management practices and antimicrobial resistance phenotype of enteric isolates from cows fecal samples. Thus, I encourage you to revise your manuscript, accordingly, taking into account all of the concerns raised by the three reviewers.

Please ensure that your figures and tables contain all of the information that is necessary to support your findings and observations. There appear to be some issues with significance and digits.

Please edit the manuscript for clarity and typos.

Please note per Reviewer 3 some important suggestions to improve the discussion of your findings.

While the concerns of the reviewers are relatively minor, this is a major revision to ensure that the original reviewers have a chance to evaluate your responses to their concerns. There are many suggestions, which I am sure will greatly improve your manuscript once addressed.

Good luck with your revision,

-joe

Reviewer 1 ·

Basic reporting

Thank you for the opportunity to read this paper. I found it interesting, well written and relevant. The background content, references and cited literature appear to be sufficient. I have a number of comments / suggestions regarding the presentation of the data and reporting of data which i address in the general comments. The results presented are relevant to the proposed hypotheses in the paper.

There is a problem in the reporting of data and significant digits, how the graphs are presented, and such throughout the paper. These are discussed later below. I encourage the authors to be mindful of every number typed/presented in the paper and their associated error reported to ensure their reporting is consistent not only from a ‘significant figures’ perspective, but meaningful. It does not, for example make sense to report tens of cows to 4 significant figures.

I also have some challenges with some of the figures presented in this submission. While I appreciate the importance of the decision tree figure(s) in the submission, much of the presentation should be compressed and more easy to read. As is, it is very difficult to absorb and will likely detract some readers, which would be unfortunate since it is very rich in data and valuable. Details are provided further below.

Experimental design

The paper straddles both data science techniques and practical questions related to dairy sciences. It does, however, raise the challenge in the extent of expertise needed to understand the submission if readers are unfamiliar with either dairy sciences or machine learning. I commend the authors attempts in trying to bridge this gap.

The research conducted on exploring the need to understand the relationships between herd metrics and bacteria is important and this research meets an unmet need in further uncovering the relationships between culled cattle and these 3 important strains. The methods in the paper were sufficiently described such that the result could be replicated, however, the authors have not made their code available for testing / review on public platforms (eg., GitHub). While it is understandable that the data itself may not be sharable, the primary codes/routines could be easily shared and I would encourage the authors to do so.

Validity of the findings

The findings in the paper appear to make sense and the discussion/conclusions drawn from them are largely reasonable. There are a few exceptions to this noted below.

Additional comments

General Comments to Author

I provide a line-by-line assessment of this paper.

Title: consider a colon instead of a comma before ‘machine learning’.

Abstract:
L54: “In contrast, …” this sentence could be written more clearly. Also, the phrase “may reflect economic decisions…” needs to be used carefully. We do not know what motivates farmers to cull. This reason (economic) may be just one of several.

Introduction:
L66: “… a serious threat…” authors should consider adding what that threat actually is for readers unfamiliar with this problem/challenge.
L75: .. 31.X% should be used if using a CI of 26.0-35.0 (specify the value X).

Materials & Methods
L161: I don’t believe MIC was defined yet.
L190-193: It is not clear to me that there are two classifications defined here, which makes the “Fifth” classification defined in the following sentence unclear. It would be helpful to write out the 2 specific classifications.
L216: it would be helpful to know the grid search algorithm used here. It *could* be confused with a manual grid search.
L228: Consider the precise use of ‘Gini Impurity’ to completely avoid possible confusion for readers unfamiliar with this metric.

Results
L239: Here is an example of inconsistent sig figs. (24% +/- 2.81, should be 24.X % +/- 2.8). I encourage the authors to be mindful of significant figures and their reporting throughout the entire text. For example, L 242 reports 31.09% +/- 3.00 but I highly doubt one can be accurate to within 3 sig figs in the error here. Same challenges in L265 onwards.

L296: It would be helpful fo the reader to know if values of 0.47 to 0.74 (and their precision) are acceptable or unacceptable ranges of sensitivity in this (diary science) context. In some medical fields 0.7 would be considered ‘poor’ whereas in other areas they might be considered ‘fair’.

L304: insert “Gini”
L308: It may be helpful to have defined entropy score previously in the methods section along with the Gini impurity.

L393: Sentence requires “The” (or something else before “Algorithms”)
L412: This sentence could be controversial and it may be wise to state that culling due to economic factors is perhaps but one of multiple reasons for culling.

Conclusions: no comments

References: appear to be in order

Figures
Figure1: Caption and Figure – no comments
Figure 2: This figure is important and it would be in the authors best interest to make this as easy as possible to digest. Some suggestions include color coding instead of text to define class, use heat-map instead of reporting actual Gini, replace ‘No resistance’ with “NR”, use ‘n’ instead of ‘samples’, and use the same size box if possible. Also, there are some issues with sig.figs (eg., why is Gini 0.64 in one box and 0.798 in another?).

Figure3: again, sig figs for values should be consistent.

Figure 4: no comments.

Figure 5: I am not sure it is necessarily to provide 10 plots here, and I think it may be more instructive to see these sensitivities in relations to other features. It should be possible to, for example, provide multiple plots on a single axes since values/ranges in y are comparable for many investigated.

Figure 6 and 7: no comments

Tables:
Table 1: no comments, nice work!
Remaining tables – no. comments

Reviewer 2 ·

Basic reporting

The manuscript was wrote with an acceptable english. The literature references and the introduction is clear to understand the manuscript. The figure and table was sufficient to understand the result in the manuscript.

Experimental design

I consider that method used was sufficient to complete aims of the manuscript.

Validity of the findings

no comment

Additional comments

The manuscript describe the association of herd management practices and antimicrobial resistance phenotype of Salmonella, E. coli and Enterococcus isolates from cows (fecal samples). The results and the methods used in the paper are clear. I consider that the paper is very interesting.

·

Basic reporting

1)The greatest strength of this article is the clear description of the 3 decision tree algorithms, and its application in classifying/ranking the herd and individual cow features to explain the shedding of multi/antimicrobial drug resistance organisms from cull cows, a potential transmission risk to humans.
2)The language used is crisp and clear to me. Various methodologies - the data collection using surveys – fecal sample collection - antimicrobial resistance profiling - machine learning techniques used, are explained appropriately. Readers in our field should be able to grasp the key findings from the neatly organized figures and tables, provided they grasp the tree classification methodologies used. Introduction, background, and references used are appropriate.

Experimental design

1)Sub-sections a) farm surveys and sampling, bacteriological culture, and antimicrobial susceptibility testing are described with appropriate details, to ensure the repeatability of the experiments.
2)Having 5 different classifications for drug resistance is appropriate to discern both herd and individual cow level features contributing to MDR/AMR/No resistance.
3)Hyper-tuning (grid-search) ensuring the selection of appropriate criteria. Influential metrics like gini, SHAP are discussed in sufficient detail and can be easily perceived by the reader.
4)All the relevant features pertaining to decision tree, random forest and gradient boosting have been harnessed meticulously, in describing the plausible herd management and individual cow factors affecting MDR organism’s shedding.

Validity of the findings

1)I think the authors should discuss the reasons (even though it is conjecture) why high rolling herd average milk production is the number one feature explaining MDR, when decision tree is used. I would think that higher rolling herd averages are associated with larger herd sizes, which indeed provide an opportunity for resistant bacteria to spread profusely. Do you agree, else provide a reason for this.
2)Similarly higher culling %, culling frequency will naturally group those herds as high risk herds for MDR bacterial shedding. Higher the culling %  higher the probability to find MDR. Discuss how this will be helpful in formulating policies for the data-informed risk assessment tool you talk about in conclusion.
3)It is counter-intuitive to me, when you say that number culled per day is negatively associated with MDR shedding, while culled % is positively associated. Isn’t number culled per day used to calculate culling % on a farm. Help me out here. So what is your suggestion to farms with regards to culling policy and MDR risk profiling? Cull lesser % of herd?

Additional comments

1)Consider shortening the title of the manuscript. Do you need to mention the fecal samples were collected from cull dairy cows?
2)Lines 72 to 75. What is a probable explanation for the 10 fold increase in Salmonella shedding prevalence between Abu Aboud et al. 2016 (3.42%) & Pereira et al. 2019 (31%)?
3)Introduction is concise and to the point. Good job.
4)Line 126. Introduce your abbreviation for extra-label drug use (ELDU), here.
5)Line 208-210. So models RF and GB prevented overfitting (limitation of DTC), isn’t it?
6)Line 307. Which tables represent the trees generated by all the 5 models?
7)Line 321. Did you mean collectively instead of separately? Figure 3 description from “.. cal” is missing.
8)Line 328-330 – Chalk method seems to be relevant feature in Salmonella too?
9)Legends for figures 5 and 7 should be larger
10)What is the standard error for the partial dependence features shown in figures 5 and 7? Will it be affected by the number of herds (6) included in your study?

---

## Round 0.2 · Minor Revisions

Dear Dr. Pandit and colleagues:

Thanks for revising your manuscript. One reviewer is satisfied with your revision, however, another still has some reservations (I consider these important to address). It appears that changes to several figures were made in haste e.g., incomplete sentences inserted). Figure 2 would greatly benefit from some graphic artist support, as this is a very important figure for readers. There are other minor suggestions by reviewer 1 that stand to greatly improve your work.

Please address these ASAP so we may move towards acceptance of your work.

Best,

-joe

Reviewer 1 ·

Basic reporting

Thank you for the opportunity to read this resubmitted paper.

- The resubmission reads well, literature cited appears up to date, the structure of the paper is appropriate.
- I commend the authors for publishing their scripts for other investigators to explore, validate and test
- I commend the authors for addressing challenges with the figures. Figure 2 is much improved, but also contains a lot of ‘dead space’ between figures and the font is very small and difficult to read. I would strongly recommend the authors obtain professional support for this figure
- There remains still some challenges with presenting numerical values, such as in line 270 onwards. Can the authors confidently state errors can be reported to within 4 significant figures, eg. 50.00% ±11.18 ? Most likely errors are no more precise than 2 sig. figs. L281 reports 42.3%±9.68 which is simply wrong. I strongly recommend the authors refer to guidelines such as the ISO GUM, AIP data representation standards for reporting numerical results, or another reference and very carefully review every number presented in their findings.

Experimental design

- There are no concerns on the experimental design. Nice work!

Validity of the findings

- There are no concerns on the validity and impact of these findings. It is a nice paper and will have value in the community.

Additional comments

I provide a line-by-line response
L35: consider using a dash (-) or bracket to separate ‘machine learning algoriths’ from the 3 types you investigate
L91: Random Forest is capitalized here and not elsewhere. Would be good to be consistent.
L392: consider replacing ‘if’ with ‘whether’ and pluralizing ‘reveal’
L394: sentence is awkward with comma after Athough. Consider re-writing.
L399: Consider replacing “Machine learning algorithms” with ‘The three machine learning algorithms tested in this study”
L410: “milk”? production?
L414: An increased percent cull and rate would reflect…?
L416: Consider inserting “total” number culled.
L419: needs a period to close sentence.
L423: Please rewrite/edit the sentence more clearly.
L430: Please rewrite/edit this sentence more clearly.

·

Basic reporting

Not Applicable

Experimental design

Not Applicable

Validity of the findings

Not Applicable

---

## Round 0.3 · accepted · Accept

Dear Dr. Pandit and colleagues:

Thanks for revising your manuscript based on the concerns raised by the reviewers. I now believe that your manuscript is suitable for publication. Congratulations! I look forward to seeing this work in print, and I anticipate it being an important resource for groups studying herd management practices and antimicrobial resistance phenotype of enteric isolates from cows fecal samples. Thanks again for choosing PeerJ to publish such important work.

Best,

-joe